# Efficient Structured Matrix Rank Minimization

**Adams Wei Yu[†], Wanli Ma[†], Yaoliang Yu[†], Jaime G. Carbonell[†], Suvrit Sra[‡]**
School of Computer Science, Carnegie Mellon University[†]
Max Planck Institute for Intelligent Systems[‡]
{weiyu, mawanli, yaoliang, jgc}@cs.cmu.edu, suvrit@tuebingen.mpg.de

## Abstract

We study the problem of finding structured low-rank matrices using nuclear norm regularization where the structure is encoded by a linear map. In contrast to most known approaches for linearly structured rank minimization, we do not (a) use the full SVD; nor (b) resort to augmented Lagrangian techniques; nor (c) solve linear systems per iteration. Instead, we formulate the problem differently so that it is amenable to a generalized conditional gradient method, which results in a practical improvement with low per iteration computational cost. Numerical results show that our approach significantly outperforms state-of-the-art competitors in terms of running time, while effectively recovering low rank solutions in stochastic system realization and spectral compressed sensing problems.

## 1 Introduction

Many practical tasks involve finding models that are both simple and capable of explaining noisy observations. The model complexity is sometimes encoded by the rank of a parameter matrix, whereas physical and system level constraints could be encoded by a specific matrix structure. Thus, rank minimization subject to structural constraints has become important to many applications in machine learning, control theory, and signal processing [10, 22]. Applications include collaborative filtering [23], system identification and realization [19, 21], multi-task learning [28], among others.

The focus of this paper is on problems where in addition to being low-rank, the parameter matrix must satisfy additional linear structure. Typically, this structure involves Hankel, Toeplitz, Sylvester, Hessenberg or circulant matrices [4, 11, 19]. The linear structure describes interdependencies between the entries of the estimated matrix and helps substantially reduce the degrees of freedom.

As a concrete example consider a linear time-invariant (LTI) system where we are estimating the parameters of an autoregressive moving-average (ARMA) model. The order of this LTI system, i.e., the dimension of the latent state space, is equal to the rank of a Hankel matrix constructed by the process covariance [20]. A system of lower order, which is easier to design and analyze, is usually more desirable. The problem of minimum order system approximation is essentially a structured matrix rank minimization problem. There are several other applications where such linear structure is of great importance—see e.g., [11] and references therein. Furthermore, since (enhanced) structured matrix completion also falls into the category of rank minimization problems, the results in our paper can as well be applied to specific problems in spectral compressed sensing [6], natural language processing [1], computer vision [8] and medical imaging [24].

Formally, we study the following (block) structured rank minimization problem:

$$\min_y \quad \tfrac{1}{2}\|\mathcal{A}(y) - b\|_\mathsf{F}^2 + \mu \cdot \mathrm{rank}(\mathcal{Q}_{m,n,j,k}(y)). \tag{1}$$

Here, $y = (y_1, ..., y_{j+k-1})$ is an $m \times n(j+k-1)$ matrix with $y_t \in \mathbb{R}^{m \times n}$ for $t = 1, ..., j+k-1$, $\mathcal{A} : \mathbb{R}^{m \times n(j+k-1)} \to \mathbb{R}^p$ is a linear map, $b \in \mathbb{R}^p$, $\mathcal{Q}_{m,n,j,k}(y) \in \mathbb{R}^{mj \times nk}$ is a structured matrix whose elements are linear functions of $y_t$'s, and $\mu > 0$ controls the regularization. Throughout this paper, we will use $M = mj$ and $N = nk$ to denote the number of rows and columns of $\mathcal{Q}_{m,n,j,k}(y)$.

Problem (1) is in general NP-hard [21] due to the presence of the rank function. A popular approach to address this issue is to use the nuclear norm $\|\cdot\|_*$, i.e., the sum of singular values, as a convex surrogate for matrix rank [22]. Doing so turns (1) into a convex optimization problem:

$$\min_y \tfrac{1}{2}\|\mathcal{A}(y) - b\|_{\mathsf{F}}^2 + \mu \cdot \|\mathcal{Q}_{m,n,j,k}(y)\|_*. \tag{2}$$

Such a relaxation has been combined with various convex optimization procedures in previous work, e.g., interior-point approaches [17, 18] and first-order alternating direction method of multipliers (ADMM) approaches [11]. However, such algorithms are computationally expensive. The cost per iteration of an interior-point method is no less than $O(M^2 N^2)$, and that of typical proximal and ADMM style first-order methods in [11] is $O(\min(N^2 M, NM^2))$; this high cost arises from each iteration requiring a full Singular Value Decomposition (SVD). The heavy computational cost of these methods prevents them from scaling to large problems.

**Contributions.** In view of the efficiency and scalability limitations of current algorithms, the key contributions of our paper are as follows.

- We formulate the structured rank minimization problem differently, so that we still find low-rank solutions consistent with the observations, but substantially more scalably.
- We customize the generalized conditional gradient (GCG) approach of Zhang et al. [27] to our new formulation. Compared with previous first-order methods, the cost per iteration is $O(MN)$ (linear in the data size), which is substantially lower than methods that require full SVDs.
- Our approach maintains a convergence rate of $O\left(\frac{1}{\epsilon}\right)$ and thus achieves an overall complexity of $O\left(\frac{MN}{\epsilon}\right)$, which is by far the lowest in terms of the dependence of $M$ or $N$ for general structured rank minimization problems. It also empirically proves to be a state-of-the-art method for (but clearly *not* limited to) stochastic system realization and spectral compressed sensing.

We note that following a GCG scheme has another practical benefit: the rank of the intermediate solutions starts from a small value and then gradually increases, while the starting solutions obtained from existing first-order methods are always of high rank. Therefore, GCG is likely to find a low-rank solution faster, especially for large size problems.

**Related work.** Liu and Vandenberghe [17] adopt an interior-point method on a reformulation of (2), where the nuclear norm is represented via a semidefinite program. The cost of each iteration in [17] is no less than $O(M^2 N^2)$. Ishteva et al. [15] propose a local optimization method to solve the weighted structured rank minimization problem, which still has complexity as high as $O(N^3 M r^2)$ per iteration, where $r$ is the rank. This high computational cost prevents [17] and [15] from handling large-scale problems. In another recent work, Fazel et al. [11] propose a framework to solve (2). They derive several primal and dual reformulations for the problem, and propose corresponding first-order methods such as ADMM, proximal-point, and accelerated projected gradient. However, each iteration of these algorithms involves a full SVD of complexity $O(\min(M^2 N, N^2 M))$, making it hard to scale them to large problems. Signoretto et al. [25] reformulate the problem to avoid full SVDs by solving an equivalent nonconvex optimization problem via ADMM. However, their method requires subroutines to solve linear equations per iteration, which can be time-consuming for large problems. Besides, there is no guarantee that their method will converge to the global optimum.

The conditional gradient (CG) (a.k.a. Frank-Wolfe) method was proposed by Frank and Wolfe [12] to solve constrained problems. At each iteration, it first solves a subproblem that minimizes a linearized objective over a compact constraint set and then moves toward the minimizer of the cost function. CG is efficient as long as the linearized subproblem is easy to solve. Due to its simplicity and scalability, CG has recently witnessed a great surge of interest in the machine learning and optimization community [16]. In another recent strand of work, CG was extended to certain regularized (non-smooth) problems as well [3, 13, 27]. In the following, we will show how a generalized CG method can be adapted to solve the structured matrix rank minimization problem.

## 2 Problem Formulation and Approach

In this section we reformulate the structured rank minimization problem in a way that enables us to apply the generalized conditional gradient method, which we subsequently show to be much more efficient than existing approaches, both theoretically and experimentally. Our starting point is that in most applications, we are interested in finding a "simple" model that is consistent with

the observations, but the problem formulation itself, such as (2), is only an intermediate means, hence it need not be fixed. In fact, when formulating our problem we can and we should take the computational concerns into account. We will demonstrate this point first.

## 2.1 Problem Reformulation

The major computational difficulty in problem (2) comes from the linear transformation $\mathcal{Q}_{m,n,j,k}(\cdot)$ inside the trace norm regularizer. To begin with, we introduce a new matrix variable $X \in \mathbb{R}^{mj \times nk}$ and remove the linear transformation by introducing the following linear constraint

$$\mathcal{Q}_{m,n,j,k}(y) = X. \tag{3}$$

For later use, we partition the matrix $X$ into the block form

$$X := \begin{bmatrix} x_{11} & x_{12} & \cdots & x_{1k} \\ x_{21} & x_{22} & \cdots & x_{2k} \\ \vdots & \vdots & & \vdots \\ x_{j1} & x_{j2} & \cdots & x_{jk} \end{bmatrix} \quad \text{with } x_{il} \in \mathbb{R}^{m \times n} \text{ for } i = 1, ..., j, \ l = 1, ..., k. \tag{4}$$

We denote by $x := \text{vec}(X) \in \mathbb{R}^{mjk \times n}$ the vector obtained by stacking the columns of $X$ blockwise, and by $X := \text{mat}(x) \in \mathbb{R}^{mj \times nk}$ the reverse operation. Since $x$ and $X$ are merely different re-orderings of the same object, we will use them interchangeably to refer to the same object.

We observe that any linear (or slightly more generally, affine) structure encoded by the linear transformation $\mathcal{Q}_{m,n,j,k}(\cdot)$ translates to linear constraints on the elements of $X$ (such as the sub-blocks in (4) satisfying say $x_{12} = x_{21}$), which can be represented as linear equations $Bx = 0$, with an appropriate matrix $B$ that encodes the structure of $\mathcal{Q}$. Similarly, the linear constraint in (3) that relates $y$ and $X$, or equivalently $x$, can also be written as the linear constraint $y = Cx$ for a suitable recovery matrix $C$. Details on constructing matrix $B$ and $C$ can be found in the appendix. Thus, we reformulate (2) into

$$\min_{x \in \mathbb{R}^{mjk \times n}} \quad \tfrac{1}{2} \|\mathcal{A}(Cx) - b\|_{\mathsf{F}}^2 + \mu \|X\|_* \tag{5}$$

$$\text{s.t.} \quad Bx = 0. \tag{6}$$

The new formulation (5) is still computationally inconvenient due to the linear constraint (6). We resolve this difficulty by applying the penalty method, i.e., by placing the linear constraint into the objective function after composing with a penalty function such as the squared Frobenius norm:

$$\min_{x \in \mathbb{R}^{mjk \times n}} \quad \tfrac{1}{2} \|\mathcal{A}(Cx) - b\|_{\mathsf{F}}^2 + \tfrac{\lambda}{2} \|Bx\|_{\mathsf{F}}^2 + \mu \|X\|_*. \tag{7}$$

Here $\lambda > 0$ is a penalty parameter that controls the inexactness of the linear constraint. In essence, we turn (5) into an *unconstrained* problem by giving up on satisfying the linear constraint exactly. We argue that this is a worthwhile trade-off for (i) By letting $\lambda \uparrow \infty$ and following a homotopy scheme the constraint can be satisfied asymptotically; (ii) If exactness of the linear constraint is truly desired, we could always post-process each iterate by projecting to the constraint manifold using $C_{\text{proj}}$ (see appendix); (iii) As we will show shortly, the potential computational gains can be significant, enabling us to solve problems at a scale which is not achievable previously. Therefore, in the sequel we will focus on solving (7). After getting a solution for $x$, we recover the original variable $y$ through the linear relation $y = Cx$. As shown in our empirical studies (see Section 3), the resulting solution $\mathcal{Q}_{m,n,j,k}(y)$ indeed enjoys the desirable low-rank property even with a moderate penalty parameter $\lambda$. We next present an efficient algorithm for solving (7).

## 2.2 The Generalized Conditional Gradient Algorithm

Observing that the first two terms in (7) are both continuously differentiable, we absorb them into a common term $f$ and rewrite (7) in the more familiar compact form:

$$\min_{X \in \mathbb{R}^{mj \times nk}} \phi(X) := f(X) + \mu \|X\|_*, \tag{8}$$

which readily fits into the framework of the generalized conditional gradient (GCG) [3, 13, 27]. In short, at each iteration GCG successively linearizes the smooth function $f$, finds a descent direction by solving the (convex) subproblem

$$Z_k \in \arg \min_{\|Z\|_* \leq 1} \langle Z, \nabla f(X_{k-1}) \rangle, \tag{9}$$

---

**Algorithm 1** Generalized Conditional Gradient for Structured Matrix Rank Minimization

---

1: Initialize $U_0$, $V_0$;
2: **for** $k = 1, 2, ...$ **do**
3:    $(u_k, v_k) \leftarrow$ top singular vector pair of $- \nabla f(U_{k-1}V_{k-1})$;
4:    set $\eta_k \leftarrow 2/(k+1)$, and $\theta_k$ by (13);
5:    $U_{\text{init}} \leftarrow (\sqrt{1 - \eta_k}U_{k-1}, \sqrt{\theta_k}u_k)$; $V_{\text{init}} \leftarrow (\sqrt{1 - \eta_k}V_{k-1}, \sqrt{\theta_k}v_k)$;
6:    $(U_k, V_k) \leftarrow \arg\min \psi(U, V)$ using initializer $(U_{\text{init}}, V_{\text{init}})$;
7: **end for**

---

and then takes the convex combination $X_k = (1-\eta_k)X_{k-1} + \eta_k(\alpha_k Z_k)$ with a suitable step size $\eta_k$ and scaling factor $\alpha_k$. Clearly, the efficiency of GCG heavily hinges on the efficacy of solving the subproblem (9). In our case, the minimal objective is simply the matrix spectral norm of $-\nabla f(X_k)$ and the minimizer can be chosen as the outer product of the top singular vector pair. Both can be computed essentially in linear time $O(MN)$ using the Lanczos algorithm [7].

To further accelerate the algorithm, we adopt the local search idea in [27], which is based on the variational form of the trace norm [26]:

$$\|X\|_* = \tfrac{1}{2}\min\{\|U\|_{\mathsf{F}}^2 + \|V\|_{\mathsf{F}}^2 : X = UV\}. \tag{10}$$

The crucial observation is that (10) is separable and smooth in the factor matrices $U$ and $V$, although not jointly convex. We alternate between the GCG algorithm and the following nonconvex auxiliary problem, trying to get the best of both ends:

$$\min_{U,V} \ \psi(U, V), \ \text{ where } \ \psi(U, V) = f(UV) + \tfrac{\mu}{2}(\|U\|_{\mathsf{F}}^2 + \|V\|_{\mathsf{F}}^2). \tag{11}$$

Since our smooth function $f$ is quadratic, it is easy to carry out a line search strategy for finding an appropriate $\alpha_k$ in the convex combination $X_{k+1} = (1-\eta_k)X_k + \eta_k(\alpha_k Z_k) =: (1-\eta_k)X_k + \theta_k Z_k$, where

$$\theta_k = \arg\min_{\theta \geq 0} h_k(\theta) \tag{12}$$

is the minimizer of the function (on $\theta \geq 0$)

$$h_k(\theta) := f((1-\eta_k)X_k + \theta Z_k) + \mu(1-\eta_k)\|X_k\|_* + \mu\theta. \tag{13}$$

In fact, $h_k(\theta)$ upper bounds the objective function $\phi$ at $(1-\eta_k)X_k + \theta Z_k$. Indeed, using convexity,

$$\begin{aligned}
\phi((1-\eta_k)X_k + \theta Z_k) &= f((1-\eta_k)X_k + \theta Z_k) + \mu\|(1-\eta_k)X_k + \theta Z_k\|_* \\
&\leq f((1-\eta_k)X_k + \theta Z_k) + \mu(1-\eta_k)\|X_k\|_* + \mu\theta\|Z_k\|_* \\
&\leq f((1-\eta_k)X_k + \theta Z_k) + \mu(1-\eta_k)\|X_k\|_* + \mu\theta \quad (\text{as } \|Z_k\|_* \leq 1) \\
&= h_k(\theta).
\end{aligned}$$

The reason to use the upper bound $h_k(\theta)$, instead of the true objective $\phi((1-\eta_k)X_k + \theta Z_k)$, is to avoid evaluating the trace norm, which can be quite expensive. More generally, if $f$ is not quadratic, we can use the quadratic upper bound suggested by the Taylor expansion. It is clear that $\theta_k$ in (12) can be computed in closed-form.

We summarize our procedure in Algorithm 1. Importantly, we note that the algorithm *explicitly* maintains a low-rank factorization $X = UV$ throughout the iteration. In fact, we never need the product $X$, which is a crucial step in reducing the memory footage for large applications. The maintained low-rank factorization also allows us to more efficiently evaluate the gradient and its spectral norm, by carefully arranging the multiplication order. Finally, we remark that we need not wait until the auxiliary problem (11) is fully solved; we can abort this local procedure whenever the gained improvement does not match the devoted computation. For the convergence guarantee we establish in Theorem 1 below, only the descent property $\psi(U_kV_k) \leq \psi(U_{k-1}V_{k-1})$ is needed. This requirement can be easily achieved by evaluating $\psi$, which, unlike the original objective $\phi$, is computationally cheap.

## 2.3 Convergence analysis

Having presented the generalized conditional gradient algorithm for our structured rank minimization problem, we now analyze its convergence property. We need the following standard assumption.

**Assumption 1** *There exists some norm $\| \cdot \|$ and some constant $L > 0$, such that for all $A, B \in \mathbb{R}^{N \times M}$ and $\eta \in (0, 1)$, we have*

$$f((1 - \eta)A + \eta B) \leq f(A) + \eta \langle B - A, \nabla f(A) \rangle + \frac{L\eta^2}{2} \|B - A\|^2.$$

Most standard loss functions, such as the quadratic loss we use in this paper, satisfy Assumption 1.

We are ready to state the convergence property of Algorithm 1 in the following theorem. To make the paper self-contained, we also reproduce the proof in the appendix.

**Theorem 1** *Let Assumption 1 hold, $X$ be arbitrary, and $X_k$ be the $k$-th iterate of Algorithm 1 applied on the problem* (7), *then we have*

$$\phi(X_k) - \phi(X) \leq \frac{2C}{k + 1}, \tag{14}$$

*where $C$ is some problem dependent absolute constant.*

Thus for any given accuracy $\epsilon > 0$, Algorithm 1 will output an $\epsilon$-approximate (in the sense of function value) solution in at most $O(1/\epsilon)$ steps.

### 2.4 Comparison with existing approaches

We briefly compare the efficiency of Algorithm 1 with the state-of-the-art approaches; more thorough experimental comparisons will be conducted in Section 3 below. The per-step complexity of our algorithm is dominated by the subproblem (9) which requires only the leading singular vector pair of the gradient. Using the Lanczos algorithm this costs $O(MN)$ arithmetic operations [16], which is significantly cheaper than the $O(\min(M^2 N, N^2 M))$ complexity of [11] (due to their need of full SVD). Other approaches such as [25] and [17] are even more costly.

## 3 Experiments

In this section, we present empirical results using our algorithms. Without loss of generality, we focus on two concrete structured rank minimization problems: (i) stochastic system realization (SSR); and (ii) 2-D spectral compressed sensing (SCS). Both problems involve minimizing the rank of two different structured matrices. For SSR, we compare different first-order methods to show the speedups offered by our algorithm. In the SCS problem, we show that our formulation can be generalized to more complicated linear structures and effectively recover unobserved signals.

### 3.1 Stochastic System Realization

**Model.** The SSR problem aims to find a minimal order autoregressive moving-average (ARMA) model, given the observation of noisy system output [11]. As a discrete linear time-invariant (LTI) system, an AMRA process can be represented by the following state-space model

$$s_{t+1} = Ds_t + Eu_t, \;\; z_t = Fs_t + u_t, \;\; t = 1, 2, ..., T, \tag{15}$$

where $s_t \in \mathbb{R}^r$ is the hidden state variable, $u_t \in \mathbb{R}^n$ is driving white noise with covariance matrix $G$, and $z_t \in \mathbb{R}^n$ is the system output that is observable at time $t$. It has been shown in [20] that the system order $r$ equals the rank of the block-Hankel matrix (see appendix for definition) constructed by the exact process covariance $y_i = \mathbb{E}(z_t z_{t+i}^T)$, provided that the number of blocks per column, $j$, is larger than the actual system order. Determining the rank $r$ is the key to the whole problem, after which, the parameters $D, E, F, G$ can be computed easily [17, 20]. Therefore, finding a low order system is equivalent to minimizing the rank of the Hankel matrix above, while remaining consistent with the observations.

**Setup.** The meaning of the following parameters can be seen in the text after E.q. (1). We follow the experimental setup of [11]. Here, $m = n$, $p = n \times n(j + k - 1)$, while $v = (v_1, v_2, ..., v_{j+k-1})$ denotes the empirical process covariance calculated as $v_i = \frac{1}{T} \sum_{t=1}^{T-i} z_{t+i} z_t^T$, for $1 \leq i \leq k$ and 0 otherwise. Let $w = (w_1, w_2, ..., w_{j+k-1})$ be the observation matrix, where the $w_i$ are all 1's for $1 \leq i \leq k$, indicating the whole block of $v_i$ is observed, and all 0's otherwise (for unobserved

blocks). Finally, $\mathcal{A}(y) = \text{vec}(w \circ y)$, $b = \text{vec}(w \circ v)$, $\mathcal{Q}(y) = H_{n,n,j,k}(y)$, where $\circ$ is the element-wise product and is $H_{n,n,j,k}(\cdot)$ the Hankel matrix (see Appendix for the corresponding $B$ and $C$).

**Data generation.** Each entry of the matrices $D \in \mathbb{R}^{r \times r}$, $E \in \mathbb{R}^{r \times n}$, $F \in \mathbb{R}^{n \times r}$ is sampled from a Gaussian distribution $N(0,1)$. Then they are normalized to have unit nuclear norm. The initial state vector $s_0$ is drawn from $N(0, I_r)$ and the input white noise $u_t$ from $N(0, I_n)$. The measurement noise is modeled by adding an $\sigma\xi$ term to the output $z_t$, so the actual observation is $\bar{z}_t = z_t + \sigma\xi$, where each entry of $\xi \in \mathbb{R}^n$ is a standard Gaussian noise, and $\sigma$ is the noise level. Throughout this experiment, we set $T = 1000$, $\sigma = 0.05$, the maximum iteration limit as 100, and the stopping criterion as $\|x_{k+1} - x_k\|_{\mathsf{F}} < 10^{-3}$ or $\frac{|\phi_{k+1} - \phi_k|}{|\min(\phi_{k+1}, \phi_k)|} < 10^{-3}$. The initial iterate is a matrix of all ones.

**Algorithms.** We compare our approach with the state-of-the-art competitors, i.e., the first-order methods proposed in [11]. Other methods, such as those in [15, 17, 25] suffer heavier computation cost per iteration, and are thus omitted from comparison. Fazel et al. [11] aim to solve either the primal or dual form of problem (2), using primal ADMM (PADMM), a variant of primal ADMM (PADMM2), a variant of dual ADMM (DADMM2), and a dual proximal point algorithm (DPPA). As for solving (7), we implemented generalized conditional gradient (GCG) and its local search variant (GCGLS). We also implemented the accelerated projected gradient with singular value threshold-ing (APG-SVT) to solve (8) by adopting the FISTA [2] scheme. To fairly compare both lines of methods for different formulations, in each iteration we track their objective values, the squared loss $\frac{1}{2}\|\mathcal{A}(Cx) - b\|_{\mathsf{F}}^2$ (or $\frac{1}{2}\|\mathcal{A}(y) - b\|_{\mathsf{F}}^2$), and the rank of the Hankel matrix $H_{m,n,j,k}(y)$. Since square loss measures how well the model fits the observations, and the Hankel matrix rank approximates the system order, comparison of these quantities obtained by different methods is meaningful.

**Result 1: Efficiency and Scalability.** We compare the performance of different methods on two sizes of problems, and the result is shown in Figure 2. The most important observation is, our approach GCGLS/GCG significantly outperform the remaining competitors in term of running time. It is easy to see from Figure 2(a) and 2(b) that both the objective value and square loss by GCGLS/GCG drop drastically within a few seconds and is at least one order of magnitude faster than the runner-up competitor (DPPA) to reach a stable stage. The rest of baseline methods cannot even approach the minimum values achieved by GCGLS/GCG within the iteration limit. Figure 2(d) and 2(e) show that such advantage is amplified as size increases, which is consistent with the theoretical finding. Then, not surprisingly, we observe that the competitors become even slower if the problem size continues growing. Hence, we only test the scalability of our approach on larger sized problems, with the running time reported in Figure 1. We can see that the running time of GCGLS grows linearly w.r.t. the size $MN$, again consistent with previous analysis.

**Result 2: Rank of solution.** We also report the rank of $H_{n,n,j,k}(y)$ versus the running time in Figure 2(c) and 2(f), where $y = Cx$ if we solve (2) or $y$ directly comes from the solution of (7). The rank is computed as the number of singular values larger than $10^{-3}$. For the GCGLS/GCG, the iterate starts from a low rank estimation and then gradually approaches the true one. However, for other competitors, the iterate first jumps to a full rank matrix and the rank of later iterate drops gradually. Given that the solution is intrinsically of low rank, GCGLS/GCG will probably find the desired one more efficiently. In view of this, the working memory of GCGLS is usually much smaller than the competitors, as it uses two low rank matrices $U, V$ to represent but never materialize the solution until necessary.

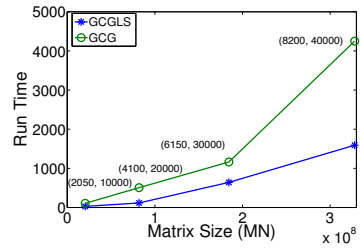

Figure 1: Scalability of GCGLS and GCG. The size $(M, N)$ is labeled out.

## 3.2 Spectral Compressed Sensing

In this part we apply our formulation and algorithm to another application, spectral compressed sensing (SCS), a technique that has by now been widely used in digital signal processing applications [6, 9, 29]. We show in particular that our reformulation (7) can effectively and rapidly recover partially observed signals.

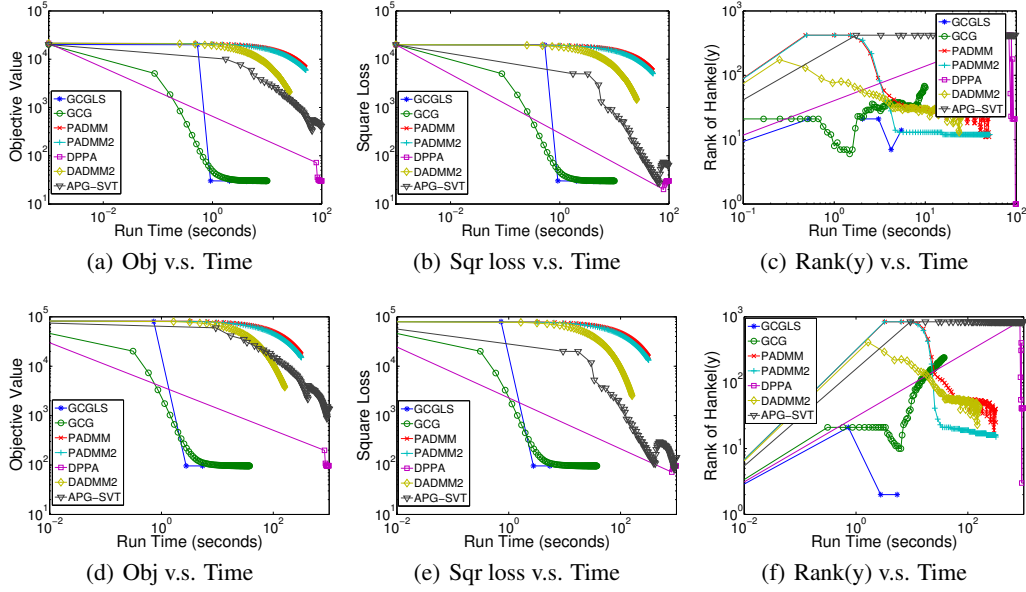

Figure 2: Stochastic System Realization problem with $j = 21, k = 100, r = 10, \mu = 1.5$ for formulation (2) and $\mu = 0.1$ for (7). The first row corresponds to the case $M = 420, N = 2000, n = m = 20,$ . The second row corresponds to the case $M = 840, N = 4000, n = m = 40$.

**Model.** The problem of spectral compressed sensing aims to recover a frequency-sparse signal from a small number of observations. The 2-D signal $Y(k,l), 0 < k \leq n_1, 0 < l \leq n_2$ is supposed to be the superposition of $r$ 2-D sinusoids of arbitrary frequencies, i.e. (in the DFT form)

$$Y(k,l) = \sum_{i=1}^{r} d_i (e^{j2\pi(kf_{1i}+lf_{2i})}) = \sum_{i=1}^{r} d_i (e^{j2\pi f_{1i}})^k (e^{j2\pi f_{2i}})^l \tag{16}$$

where $d_i$ is the amplitudes of the $i$-th sinusoid and $(f_{xi}, f_{yi})$ is its frequency.

Inspired by the conventional matrix pencil method [14] for estimating the frequencies of sinusoidal signals or complex sinusoidal (damped) signals, the authors in [6] propose to arrange the observed data into a 2-fold Hankel matrix whose rank is bounded above by $r$, and formulate the 2-D spectral compressed sensing problem into a rank minimization problem with respect to the 2-fold Hankel structure. This 2-fold structure is a also linear structure, as we explain in the appendix. Given limited observations, this problem can be viewed as a matrix completion problem that recovers a low-rank matrix from partially observed entries while preserving the pre-defined linear structure. The trace norm heuristic for rank $(\cdot)$ is again used here, as it is proved by [5] to be an exact method for matrix completion provided that the number of observed entries satisfies the corresponding information theoretic bound.

**Setup.** Given a partial observed signal $\overline{Y}$ with $\Omega$ as the observation index set, we adopt the formulation (7) and thus aim to solve the following problem:

$$\min_{X \in \mathbb{R}^{M \times N}} \frac{1}{2} \|P_\Omega(\text{mat}(Cx)) - P_\Omega(\overline{Y})\|_{\mathsf{F}}^2 + \frac{\lambda}{2} \|Bx\|_{\mathsf{F}}^2 + \mu \|X\|_* \tag{17}$$

where $x = \text{vec}(X)$, $\text{mat}(\cdot)$ is the inverse of the vectorization operator on $Y$. In this context, as before, $\mathcal{A} = P_\Omega$, $b = P_\Omega(\overline{Y})$, where $P_\Omega(\overline{Y})$ only keeps the entries of $\overline{Y}$ in the index set $\Omega$ and vanishes the others, $\mathcal{Q}(Y) = H_{k_1,k_2}^{(2)}(Y)$ is the two-fold Hankel matrix, and corresponding $B$ and $C$ can be found in the appendix to encode $H_{k_1,k_2}^{(2)}(Y) = X$ . Further, the size of matrix here is $M = k_1 k_2, N = (n_1 - k_1 + 1)(n_2 - k_2 + 1)$.

**Algorithm.** We apply our generalized conditional gradient method with local search (GCGLS) to solve the spectral compressed sensing problem, using the reformulation discussed above. Following

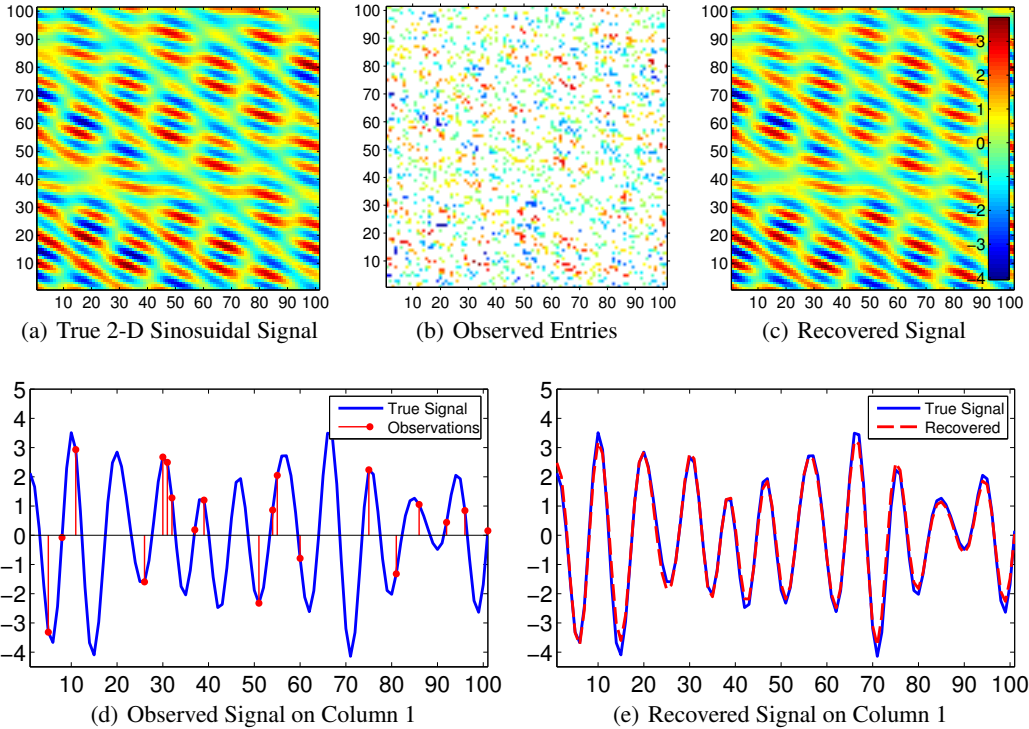

(a) True 2-D Sinosuidal Signal    (b) Observed Entries    (c) Recovered Signal

(d) Observed Signal on Column 1    (e) Recovered Signal on Column 1

Figure 3: Spectral Compressed Sensing problem with parameters $n_1 = n_2 = 101, r = 6$, solved with our GCGLS algorithm using $k_1 = k_2 = 8, \mu = 0.1$. The 2-D signals in the first row are colored by the jet colormap. The second row shows the 1-D signal extracted from the first column of the data matrix.

the experiment setup in [6], we generate a ground truth data matrix $Y \in \mathbb{R}^{101 \times 101}$ through a super-position of $r = 6$ 2-D sinusoids, randomly reveal 20% of the entries, and add i.i.d Gaussian noise with amplitude signal-to-noise ratio 10.

**Result.** The results on the SCS problem are shown in Figure 3. The generated true 2-D signal $Y$ is shown in Figure 3(a) using the jet colormap. The 20% observed entries of $Y$ are shown in Figure 3(b), where the white entries are unobserved. The signal recovered by our GCGLS algorithm is shown in Figure 3(c). Comparing with the true signal in Figure 3(a), we can see that the result of our CGCLS algorithm is pretty close to the truth. To demonstrate the result more clearly, we extract a single column as a 1-D signals for further inspection. Figure 3(d) plots the original signal (blue line) as well as the observed ones (red dot), both from the first column of the 2-D signals. In 3(e), the recovered signal is represented by the red dashed dashed curve. It matches the original signal with significantly large portion, showing the success of our method in recovering partially observed 2-D signals from noise. Since the 2-fold structure used in this experiment is more complicated than that in the previous SSR task, this experiment further validates our algorithm on more complicated problems.

## 4  Conclusion

In this paper, we address the structured matrix rank minimization problem. We first formulate the problem differently, so that it is amenable to adapt the Generalized Conditional Gradient Method. By doing so, we are able to achieve the complexity $O(MN)$ per iteration with a convergence rate $O\left(\frac{1}{\epsilon}\right)$. Then the overall complexity is by far the lowest compared to state-of-the-art methods for the structured matrix rank minimization problem. Our empirical studies on stochastic system realization and spectral compressed sensing further confirm the efficiency of the algorithm and the effectiveness of our reformulation.

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
