[Supplementary Material · nips2014_cr_appendix.pdf]

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

## Appendix: Efficient Structured Matrix Rank Minimization

**Proof of Theorem 1**

The proof follows the line of that in [27].

Fix the "competitor" $X$. We first show that

$$\phi(X_k) \leq (1 - \eta_k)\phi(X_{k-1}) + \eta_k \phi(X) + \frac{C_k \eta_k^2}{2}, \tag{18}$$

where $C_k := L \cdot \left\| \|X\|_* Z_k - X_{k-1} \right\|^2$. Indeed,

$$
\begin{aligned}
\phi(X_k) &= f(X_k) + \mu\|X_k\|_* \\
&= \min_{\theta \geq 0} f\big((1-\eta_k)X_{k-1} + \theta Z_k\big) + \mu(1-\eta_k)\|X_{k-1}\|_* + \mu\theta && [(12)] \\
&\leq f\big((1-\eta_k)X_{k-1} + \eta_k\|X\|_* Z_k\big) + \mu(1-\eta_k)\|X_{k-1}\|_* + \mu\eta_k\|X\|_* \\
&\leq f(X_{k-1}) + \eta_k \langle \|X\|_* Z_k - X_{k-1}, \nabla f(X_{k-1})\rangle + \frac{C_k\eta_k^2}{2} + \mu(1-\eta_k)\|X_{k-1}\|_* + \mu\eta_k\|X\|_* && [\text{Assumption 1}] \\
&= \phi(X_{k-1}) + \eta_k \langle \|X\|_* Z_k - X_{k-1}, \nabla f(X_{k-1})\rangle + \frac{C_k\eta_k^2}{2} - \mu\eta_k\|X_{k-1}\|_* + \mu\eta_k\|X\|_* \\
&\leq \min_{Y:\|Y\|_* \leq \|X\|_*} \phi(X_{k-1}) + \eta_k \langle Y - X_{k-1}, \nabla f(X_{k-1})\rangle + \frac{C_k\eta_k^2}{2} - \mu\eta_k\|X_{k-1}\|_* + \mu\eta_k\|X\|_* && [(9)] \\
&\leq \min_{Y:\|Y\|_* \leq \|X\|_*} \phi(X_{k-1}) + \eta_k(f(Y) - f(X_{k-1})) + \frac{C_k\eta_k^2}{2} - \mu\eta_k\|X_{k-1}\|_* + \mu\eta_k\|X\|_* && [\text{convexity of } f] \\
&= (1-\eta_k)\phi(X_{k-1}) + \eta_k \min_{Y:\|Y\|_* \leq \|X\|_*} (f(Y) + \mu\|X\|_*) + \frac{C_k\eta_k^2}{2} \\
&= (1-\eta_k)\phi(X_{k-1}) + \eta_k\phi(X) + \frac{C_k\eta_k^2}{2}.
\end{aligned}
$$

Note that we only need the local search (line 6 of Algorithm 1) to satisfy the descent property $\psi(U_k V_k) \leq \psi(U_{k-1}V_{k-1})$, so that by induction $\psi(U_k V_k) \leq \psi(U_0 V_0) = C_0$ for some constant $C_0$. Thus $\|X_k\| = \|U_k V_k\|$ is uniformly bounded, meaning that the term $C_k$ in (18) can be bounded by a universal constant $C'$ (which depends on the competitor $X$ that we fix throughout).

Therefore, we have

$$\phi(X_k) \leq (1 - \eta_k)\phi(X_{k-1}) + \eta_k \phi(X) + \frac{C'\eta_k^2}{2}, \tag{19}$$

Let $C = \max(C', \phi(X_1) - \phi(X))$. Then we show by induction that (14) holds.

1. When $k = 1$, $\phi(X_1) - \phi(X) \leq C$, (14) holds.

2. Suppose Theorem 1 holds for the $k$-th steps, i.e. $\phi(X_k) - \phi(X) \leq \frac{2C}{k+1}$, we show that it also holds for the $(k+1)$-th step. Indeed, by (19) and $\eta_{k+1} = \frac{2}{k+2}$, we have

$$
\begin{aligned}
\phi(X_{k+1}) - \phi(X) &\leq (1 - \eta_{k+1})(\phi(X_k) - \phi(X)) + \frac{C'\eta_{k+1}^2}{2} \\
&\leq \frac{k}{k+2} \cdot \frac{2C}{k+1} + \frac{2C}{(k+2)^2} \\
&= \frac{2C(k^2 + 3k + 1)}{(k+1)(k+2)^2} \\
&\leq \frac{2C}{k+2}.
\end{aligned}
$$

This concludes the proof of Theorem 1 for all steps $k$.

**Linear Structured Matrices and the corresponding Matrix $B$ and $C$**

**General Linear Matrix Structures**

In general, linear matrix structures are defined [15] as:

$$\mathcal{Q}(y) = Q_0 + \sum_{k=1}^{n_y} Q_k y_k \tag{20}$$

where $Q_k \in \mathbb{R}^{m \times n}$, $0 \le k \le n_y$ and $y \in \mathbb{R}^{n_y}$ is the given data. Let $Q_{k,i}(i \le mn)$ be the $i$'th element in $\text{vec}(Q_k)$.

We further assume that (1) $Q_0 = 0$, (2) each $Q_k$ is a (0,1)-matrix and (3) for $\forall i \le mn$, there exists at most one $k$ such that $Q_{k,i} = 1$. In other words, each element in the structured matrix $\mathcal{Q}(y)$ either equals to one element in $y$, or is 0. Most of the linear matrix structures, including block-Hankel and 2-fold Hankel used in our experiments, as well as Toeplitz, Sylvester and circulant, satisfy this assumption.

We claim that for any structure $\mathcal{Q} : \mathbb{R}^{n_y} \to \mathbb{R}^{m \times n}$ under this assumption, we can construct a "structure preserving matrix" $B$ and a "recovery matrix" $C$ such that for any $X \in \mathbb{R}^{m \times n}$

$$B\text{vec}(X) = 0 \iff \exists y \in \mathbb{R}^{n_y}, \text{ s.t. } X = \mathcal{Q}(y) \text{ and } C\text{vec}(X) = y \tag{21}$$

or in other words, $B\text{vec}(X) = 0 \Leftrightarrow X \in \text{image}(\mathcal{Q})$, where $\text{image}(\mathcal{Q}) := \{\mathcal{Q}(y) | y \in \mathbb{R}^{n_y}\}$. $B$ can be viewed as the Lagrangian of the structural constraint.

The matrix $B$ can be constructed in the following way. Let $d_k^j$ be the position of the $j$th 1 in $\text{vec}(Q_k)$. Let $|Q_k|$ be the number of 1's in $Q_k$. The structure defined above requires that for any $X \in \text{image}(\mathcal{Q})$, each pair of $(X_{d_k^j}, X_{d_k^{j+1}})$ must be equal. Since there are totally $T = \sum_{k=1}^{n_y}(|Q_k| - 1)$ such pairs, $B$ can be constructed as a $T \times mn$ sparse matrix by only assigning $B_{t,d_k^j} = 1$, $B_{t,d_k^{j+1}} = -1$ for the $t$th pair of $X_{d_k^j} = X_{d_k^{j+1}}$ constraint. In case we need to enforce some elements of $X$ to be zero, we may add more rows to $B$ with only one 1 per row at the position of the focused element.

The matrix $C$ can be constructed as a $n_y \times mn$ sparse matrix by assigning $C_{k,d_k^j} = 1/|Q_k|$ and leaving other entries 0. Note that this $C$ can be applied to arbitrary $X \in \mathbb{R}^{m \times n}$ as an orthogonal projection onto $\text{image}(\mathcal{Q})$, i.e.

$$\mathcal{Q}(C\text{vec}(X)) = \operatorname*{arg\,min}_{\hat{X} \in \text{image}(\mathcal{Q})} \|\hat{X} - X\|_{\mathsf{F}}^2$$

Thus we call this $C$ the projection matrix $C_{\text{proj}}$. One may refer to the Appendix of [15] for the proof. $C$ can be also constructed in other ways to satisfy (21), for instance, a sparser $C$ can be constructed by assigning only $C_{k,d_k^1} = 1$ for $1 \le k \le n_y$. It's easy to verify that the sparser one $C_{\text{sp}}$ is also an inverse operator of $\text{vec}(\mathcal{Q}(\cdot))$

**Example: Hankel Matrix**

In the following examples, we always use $I_k$ to denote the identity matrix of size $k \times k$ and $0_{k,j}$ to denote a zero matrix of size $k \times j$.

For a Hankel matrix of data $y \in \mathbb{R}^{j+k-1}$ parameterized by $j$ and $k$:

$$H_{j,k}(y) := \begin{bmatrix} y_1 & y_2 & \cdots & y_k \\ y_2 & y_3 & \cdots & y_{k+1} \\ \vdots & \vdots & & \vdots \\ y_j & y_{j+1} & \cdots & y_{j+k-1} \end{bmatrix} \in \mathbb{R}^{j \times k} \tag{22}$$

The Hankel structure preserving matrix $B \in \mathbb{R}^{(j-1)(k-1) \times jk}$(after rearranging the order or rows) is

$$B = \begin{bmatrix} \mathbf{P} & \mathbf{N} & \mathbf{0} & \mathbf{0} & \cdots & \mathbf{0} & \mathbf{0} \\ \mathbf{0} & \mathbf{P} & \mathbf{N} & \mathbf{0} & \cdots & \mathbf{0} & \mathbf{0} \\ \vdots & \vdots & \vdots & \vdots & \vdots & \vdots & \vdots \\ \mathbf{0} & \mathbf{0} & \mathbf{0} & \cdots & & \mathbf{0} & \mathbf{P} & \mathbf{N} \end{bmatrix} \tag{23}$$

where $\mathbf{P} = [0_{j-1,1}, I_{j-1}]$, $\mathbf{N} = [-I_{j-1}, 0_{j-1,1}]$, $\mathbf{0} = 0_{j-1,j}$. Obviously $\mathbf{P}, \mathbf{N}, \mathbf{0} \in \mathbb{R}^{(j-1)\times j}$.

For the recovery matrix $C \in \mathbb{R}^{(j+k-1)\times jk}$, we show a toy example using parameters $j = 2, k = 3$. The projection $C_{\text{proj}}$ and the sparser $C_{\text{sp}}$ are

$$
C_{\text{proj}} = \begin{bmatrix} 1 & 0 & 0 & 0 & 0 & 0 \\ 0 & 0.5 & 0.5 & 0 & 0 & 0 \\ 0 & 0 & 0 & 0.5 & 0.5 & 0 \\ 0 & 0 & 0 & 0 & 0 & 1 \end{bmatrix}, C_{\text{sp}} = \begin{bmatrix} 1 & 0 & 0 & 0 & 0 & 0 \\ 0 & 1 & 0 & 0 & 0 & 0 \\ 0 & 0 & 0 & 1 & 0 & 0 \\ 0 & 0 & 0 & 0 & 0 & 1 \end{bmatrix} \tag{24}
$$

**Example: Block-Hankel Matrix**

For the block-Hankel matrix used in the stochastic system realization experiment

$$
H_{m,n,j,k}(y) := \begin{bmatrix} y_1 & y_2 & \cdots & y_k \\ y_2 & y_3 & \cdots & y_{k+1} \\ \vdots & \vdots & & \vdots \\ y_j & y_{j+1} & \cdots & y_{j+k-1} \end{bmatrix} \in \mathbb{R}^{mj\times nk} \tag{25}
$$

where each $y_1, \ldots, y_{j+k-1}$ is a $m \times n$ data matrix. If we define $\text{vec}(\cdot)$ blockwise, we can write the matrix $B \in \mathbb{R}^{m(j-1)(k-1)\times mjk}$ in the same form as (23) where $\mathbf{P} = [0_{m(j-1),m}, I_{m(j-1)}]$, $\mathbf{N} = [-I_{m(j-1)}, 0_{m(j-1),m}]$, $\mathbf{0} = 0_{m(j-1),mj}$, $\mathbf{P}, \mathbf{N}, \mathbf{0} \in \mathbb{R}^{m(j-1)\times mj}$.

The matrix $C \in \mathbb{R}^{m(j+k-1)\times mjk}$ can be constructed from (24) by replacing each element $a$ with a block $aI_m$.

**Example: Two-Fold Hankel Matrix**

For a 2-D data matrix $Y \in \mathbb{R}^{n_1 \times n_2}$, the enhanced form $H^{(2)}_{k_1,k_2}(Y)$ with respect to the pencil parameter $k_1$ and $k_2$ is a block-Hankel matrix with $k_1 \times (n_1 - k_1 + 1)$ blocks [6]:

$$
H^{(2)}_{k_1,k_2}(Y) := \begin{bmatrix} \mathbf{Y}_1 & \mathbf{Y}_2 & \cdots & \mathbf{Y}_{n_1-k_1+1} \\ \mathbf{Y}_2 & \mathbf{Y}_3 & \cdots & \mathbf{Y}_{n_1-k_1+2} \\ \vdots & \vdots & \vdots & \vdots \\ \mathbf{Y}_{k_1} & \mathbf{Y}_{k_1+1} & \cdots & \mathbf{Y}_{n_1} \end{bmatrix} \tag{26}
$$

and each block $\mathbf{Y}_l$ ($0 < l \leq n_1$) is a (micro) Hankel matrix of size $k_2 \times (n_2 - k_2 + 1)$

$$
\mathbf{Y}_l := H_{1,1,k_2,n_2-k_2+1}(Y(l,:)) = \begin{bmatrix} Y_{l,1} & Y_{l,2} & \cdots & Y_{l,n_2-k_2+1} \\ Y_{l,2} & Y_{l,3} & \cdots & Y_{l,n_2-k_2+2} \\ \vdots & \vdots & \vdots & \vdots \\ Y_{l,k_2} & Y_{l,k_2+1} & \cdots & Y_{l,n_2} \end{bmatrix} \tag{27}
$$

Here we use $H^{(2)}$ to denote the 2-fold Hankel structure. $H^{(2)}$ has $M = k_1 k_2$ rows and $N = (n_1 - k_1 + 1)(n_2 - k_2 + 1)$ columns.

Here $B$ is a matrix with $k_1(k_2 - 1)(n_2 - k_2)(n_1 - k_1 + 1) + n_2(k_1 - 1)(n_1 - k_1)$ rows and $MN$ columns:

$$
B := \begin{bmatrix} B_1 & 0 & \cdots & 0 \\ 0 & B_1 & \cdots & 0 \\ \vdots & \vdots & \vdots & \vdots \\ 0 & 0 & \cdots & B_1 \\ \hdashline & & B_2 & \end{bmatrix} \tag{28}
$$

such that each $B_1 \in \mathbb{R}^{k_1(k_2-1)(n_2-k_2)\times k_1 k_2(n_2-k_2+1)}$ preserves the micro Hankel structure of $k_1$ blocks in one "block-wise column" of $X$ and $B_2 \in \mathbb{R}^{n_2(k_1-1)(n_1-k_1)\times MN}$ preserves the global block-Hankel structure. Both $B_1$ and $B_2$ as well as the recovery matrix $C_{\text{proj}}$ are constructed using the steps mentioned above.