[Reviews · NeurIPS 2014]

Submitted by Assigned_Reviewer_13

Summary: The paper describes an efficient optimization approach to find structured low-rank matrices.
The structure is encoded by a linear map and enforcing low rank is achieved by adding to the cost function the nuclear norm of the structured matrix.
The cost function is optimized with a generalized conditional gradient algorithm.
By using a factorization of the large structured matrix the optimization is accelerated further. This comes at the cost of having to solve a non-convex auxiliary problem, optimizing the two factor matrices. But this can be solved efficiently following ideas of Zhang et al. (NIPS2012).
A theoretical analysis indicates that the whole algorithm has a remarkably low complexity and for several sample problems it is shown that this approach is at least an order of magnitude faster than the next best competitor. It also scales well to very large problems; speed comparisons are shown for matrix sizes up to over 300M elements. An advantage is that the factor matrices start from low ranks while alternative methods tend to start from full rank.

Quality: A thorough theoretical analysis is provided for computational complexity and convergence rate that show significant improvements over alternative methods. This is also confirmed in experimental results where the method outperforms consistently the competing methods.

Clarity: Overall the paper is well written, but the material is complex and the reader has to go through the supplementary file as well as through some of the cited papers to appreciate what is happening.

Originality: The method described here combines in a creative way several ideas from previous algorithms to come up with a new solution that is surprisingly efficient.

Significance: The problem that is addressed here is of significance for various applications and the results show convincing improvements over state of the art.
Summary: The paper describes an efficient optimization approach to find structured low-rank matrices. A theoretical analysis indicates considerable improvements over alternative methods and experimental results show an order of magnitude improvement in speed over competing techniques.

Submitted by Assigned_Reviewer_15

This work considers a seemingly new, and faster, way to deal
with nuclear-norm regularised estimation of low-rank matrices.
The work is well motivated. The authors take time to carefully
explain the main contributions and how these fit in with the
existing literature.

The authors perform a reasonably simple mathematical trick to
manipulate the form of the problem. The main work appears to be
in the exploitation of this trick using their knowledge of the
conditional gradient algorithm.
Summary: A somewhat novel algrotihm is proposed to estimate low-rank matrices. The algorithm offers a speed-up compared to existing methods.

Submitted by Assigned_Reviewer_44

** I have read the authors’ response and they have provided satisfactory comments addressing my major concerns. I discuss some of these below (denoting added comments with **). I have read the other reviews and my rating of the paper hasn’t changed; I think it is a strong paper and should be accepted. I do not feel that the theoretical or experimental contribution pushes it into the oral/spotlight category.**

The paper addresses the problem of finding structured low rank matrices for the case when the structure is encoded using a linear map and nuclear norm regularization is employed. The authors propose a modified formulation of the problem which makes it amenable to application of the generalized conditional gradient method. They provide an algorithm that incorporates other methods that accelerate convergence and show that it converges. Empirical studies suggest that the algorithm is significantly faster than its competitors and can scale to much larger problems.

Strengths:

(1) The paper addresses an optimization problem that lies at the core of multiple machine learning and signal processing tasks and the authors provide a novel procedure that can identify a suitable solution much faster than state-of-the-art approaches. The empirical studies demonstrate the efficacy of the approach for two useful application problems.

Weaknesses:

(1) The authors achieve the significant reduction in computational cost by modifying the objective function so that it is no longer directly penalizes the rank of the structured matrix. The authors make three reasonable arguments for doing so shortly after stating the objective function in (7). The first two argue that the linear constraint could be satisfied asymptotically through a homotopy scheme or exactly through a projection, but the paper doesn’t explore either of these options, either empirically or theoretically. The theoretical result really addresses the convergence of the algorithm in terms of minimizing the modified objective function; it doesn’t provide a sense of the discrepancy between the identified solution and the more natural original formulation. Graphs 2(c) and 2(f) show some slightly strange behaviour in the rank of the solution as it converges. This behaviour is not discussed at all and it is not clear when the algorithm should be terminated to identify an appropriate solution.

** The authors have argued that “any formulation that achieves this goal should be sufficient for the application”. I think this is a reasonable argument, but at the moment the paper is presented as though (7) is a relaxation/approximation of (5) – and is effectively trying to find the same solution. I think it would be useful to add a sentence or two making it clear that this is an alternative utility function that targets the same overall objective (low-rank model). **
** With regard to the experimental results and the behaviour of the algorithm, it would be good to add some extra discussion either in the main paper or in the supplementary material expanding on what the authors have provided in their response” **

Comments:

Why does Theorem 1 cite [28]? Although there is a similar theorem in [28], with a similar proof approach, this doesn’t seem to be taken directly from there? What does ([28]) mean?

** The authors acknowledge that the citation should appear in the proof, after a statement that the proof technique is inspired by the method in [28]. **
Summary: The paper addresses an optimization problem that lies at the core of multiple machine learning and signal processing tasks and the authors provide a novel procedure that can identify a suitable solution much faster than state-of-the-art approaches. This is achieved by modifying the objective function and the paper would be improved if it provided more in-depth discussion and analysis of the impact of this modification.
Author Feedback
Author rebuttal: We would like to thank all the reviewers for their appreciation of our contributions! Their insightful comments and suggestions will definitely help us improve the paper. In the following, we provide responses to the comments by the reviewers.

R1: we will try to further polish the manuscript to improve its clarity. We will be grateful for any suggestions about which parts of the paper might need this the most!

R2: we thank the reviewer for the thoughtful review.

R3 expresses the concern on the approaches to force the strict linear constraint. In experiment 1, we have chosen a \lambda such that the quantity ||Bx||_F is always of the magnitude \le 10^{-3}, which indicates that the linear constraint is fairly well satisfied.

We agree with the reviewer that having a precise theoretical connection between the various formulations is of interest. But as mentioned in the paper, for many applications the main goal is to find a simple (low-rank) model consistent with the observations. Therefore, any formulation that achieves this goal should be sufficient for the application, regardless of the actual form (5), (6), or (7). To substantiate this viewpoint empirically, we see in experiment 2 that reformulation (7) does indeed effectively and rapidly recover partially observed signals, which indicates that giving up the exact linear constraint may not be detrimental.

As for Figure 2(c) and 2(f), the general trend seen therein is: the ‘y’ values of our methods GCG and GCGLS start with a low-rank estimate and might increase (with perturbation) afterwards, while the other methods start from a high-rank estimate and decrease gradually. The direct solution of our method is ‘x’, whose rank is monotonically increasing (not shown in the current draft). The transformation y=Cx and the threshold 10^{-3} (in line 311) might lead to perturbations of rank of y, but it still remains low. Therefore, in general, the proposed method can obtain low-rank solutions quite rapidly. We would definitely elaborate more on this in the final version.

Finally, just like most optimization algorithms, ours also needs efforts to make some trials to determine what may be good stopping criteria in practice, as it depends on several factors such as the problem size, the parameters \lambda, \mu, etc. For reference, we list the stopping criterion used in our experiments in line 282.

We cited [28] in front of Theorem 1 because our proof technique is inspired by theirs. But you are right; this citation belongs in the text of the proof rather than at the head of the theorem. Thanks!